# Lived experience of nutrition impact symptoms among patients undergoing chemotherapy in Ethiopia: An interpretative phenomenological analysis

Awole Seid[1,2]*, Zelalem Debebe[1], Abebe Ayelign[1], Bilal Shikur Endris[3], Mathewos Assefa[4], Ahmedin Jemal[5]

1 Center for Food Science and Nutrition, College of Natural and Computational Sciences, Addis Ababa University, Addis Ababa, Ethiopia, 2 College of Medicine and Health Sciences, Bahir Dar University, Bahir Dar, Ethiopia, 3 Department of Nutrition and Dietetics, School of Public Health, Addis Ababa University, Addis Ababa, Ethiopia, 4 Department of Oncology, School of Medicine, Addis Ababa University, Addis Ababa, Ethiopia, 5 Department of Surveillance and Health Services Research, American Cancer Society, Atlanta, Georgia, United States of America

* sawlayehu@gmail.com

## Abstract

### Background

Nutrition impact symptoms such as nausea, vomiting, and taste alterations are common side effects of chemotherapy and can lead to malnutrition. There is a paucity of data regarding the nutritional challenges faced by cancer patients, particularly in resource-limited settings. This study aimed to explore the lived experiences of nutrition impact symptoms among patients undergoing chemotherapy at a major cancer center in Ethiopia.

### Methods

An interpretative phenomenological analysis was conducted from November 11–29, 2024, involving 26 cancer patients treated at the Oncology Center of Tikur Anbesa Specialized Hospital, Addis Ababa, Ethiopia. Both data and thematic saturation were employed to determine the sample size. Participants were selected using heterogeneous sampling, and data were collected through in-depth interviews. The interviews were audio-recorded and transcribed verbatim in Amharic, followed by a contextual translation into English. The data were analyzed using an inductive thematic analysis approach with the aid of MAXQDA24 software.

### Results

Three themes were identified: symptom burden and coping, individualized food choices, and unmet nutritional support needs. Symptoms were particularly severe

---

**Data availability statement:** All relevant data are within the paper and its Supporting Information files.

**Funding:** The author(s) received no specific funding for this work.

**Competing interests:** The authors have declared that no competing interests exist.

during the early stages of treatment, disrupting typical dietary patterns and leading to physical limitations, negative emotional responses, and decreased productivity. The finding also revealed financial barriers to accessing nutritious foods, nutritional misinformation, and unsatisfactory experiences with hospital food.

## Conclusion

Symptoms vary in onset, severity, and pattern among individuals, significantly impacting their quality of life. Nutritional support is a pressing need for cancer patients. The findings underscore the critical need for dietitian-led, patient-centered nutritional interventions, along with socioeconomic support for patients undergoing chemotherapy in Ethiopia.

---

## Introduction

Cancer remains a significant public health issue globally [1,2]. In Ethiopia, GLOBOCAN estimated over 80,000 new cancer cases and more than 50,000 cancer deaths in 2022 [3]. Malnutrition affects 20–80% of cancer patients, with 20% of cancer deaths attributed to malnutrition rather than cancer itself [4,5]. Undernourishment in cancer patients results from metabolic changes due to the tumor and the side effects of treatment modalities [4,6,7]. A review indicated that individuals with cancer and poor nutritional status were less likely to complete oncologic treatments as planned and incurred higher healthcare costs than well-nourished patients [8].

Nutrition impact symptoms refer to a range of symptoms that impede food intake, compromise nutritional status, and adversely affect quality of life, including an individual's physical, mental, and social well-being [9,10]. These symptoms include nausea, vomiting, loss of appetite, constipation, diarrhea, dysphagia, mouth sores, early satiety, loss of smell, loss of taste, pain, and fatigue [11,12]. The severity of nutrition impact symptoms is associated with age, tumor location, disease stage, and treatment type [13,14]. A longitudinal study in Australia found that 79% of medical oncology patients experienced at least one nutrition impact symptom one month after starting chemotherapy [15]. In our baseline cross-sectional study, we found that 61.7% of patients in major cancer centers in Ethiopia had at least one nutrition impact symptom before chemotherapy initiation, with expected increases during and after treatment [16].

Nutritional impact symptoms significantly impair the clinical outcomes of cancer patients undergoing chemotherapy. For example, qualitative studies in Malaysia and Australia have indicated that chemotherapy-induced nausea and vomiting are distressing and affect eating, physical, emotional, and social functioning, with varying severity and antiemetic efficacy [17,18]. Another study in Taiwan indicated that nutritional care is the most prevalent unmet need (40.7%) among patients diagnosed with nasopharyngeal cancer [19]. Moreover, a recently published protocol led by the World Health Organization (WHO) highlighted that the lived experiences of cancer survivors and their families remain understudied [20]. In light of this, the WHO has launched

a global survey on the lived experiences of people affected by cancer to improve their well-being. However, the survey focused on the social, emotional, and financial impacts of cancer and inadequately explored nutritional challenges [21,22]. Furthermore, a meta-synthesis of the lived experiences of nutrition impact symptoms among patients with head and neck cancer (HNC) indicated that dysphagia, taste changes, oral mucositis, and xerostomia negatively affected patients' quality of life. However, the findings are limited to British and American contexts, making them less applicable to low-resource settings [23]. The trend of late-stage cancer diagnoses in Ethiopia, coupled with the limited availability of cancer and palliative care centers, inadequately equipped oncology facilities, the absence of nutritionists or dietitians in cancer centers, and economic barriers to accessing nutritious foods, is anticipated to exacerbate the burden of nutrition-related symptoms in patients undergoing chemotherapy.

To investigate patients' experiences of nutrition-related symptoms, a qualitative approach is appropriate, as it provides a comprehensive understanding of the phenomenon and helps create the meaning of specific behaviors within different contexts. Additionally, qualitative studies employ a constructivist paradigm, which posits understanding patients' experiences from their perspectives through researcher and participant interactions [24,25]. In our research context, using a phenomenological study allows us to gain a deeper understanding of patients' nutrition impact symptoms during chemotherapy, emotional responses, impact on quality of life, and coping strategies.

Although previous studies in Ethiopia have explored the lived experiences of cancer patients, closer examination reveals that limited attention has been given to nutritional challenges, including gastrointestinal symptoms, dietary patterns, coping methods, and experiences with hospital meals. For example, several studies have focused on the psychosocial experiences associated with cancer survival, including changes in body image resulting from hair loss or mastectomy [26–28]. Additionally, prior qualitative studies conducted in other countries often lack comprehensiveness, typically focusing on a single symptom, such as nausea or vomiting, or are performed in high-resource settings [17,18,29]. Therefore, the purpose of this study was to investigate the lived experiences of nutrition impact symptoms among patients receiving chemotherapy in Ethiopia, aiming to address qualitative research gaps within low-resource settings. This paper builds upon previous findings from a cross-sectional study that assessed the prevalence of nutrition impact symptoms in cancer patients [16]. Understanding the nutritional challenges encountered by individuals with cancer is a crucial initial step in prioritizing the management of nutrition impact symptoms and organizing oncological services to address the unmet nutritional needs of cancer patients.

## Methods

### Study setting and design

An interpretive phenomenological analysis (IPA) was conducted from November 11–29, 2024, at the Oncology Center of Tikur Anbesa Specialized Hospital (TASH), Ethiopia's pioneering cancer center found in Addis Ababa. As a referral center, it serves patients from remote areas of the country, each with diverse socio-cultural practices and food preferences.

The use of IPA allowed us for an examination of participants' lived experiences within their socio-demographic, socio-cultural, and economic contexts. The study aimed to understand not only what participants experienced but also why they made certain decisions and took specific actions [30]. This approach enabled the exploration of the core components and essence of participants' experiences beyond mere descriptions [31,32].

### Study participants and selection methods

The inclusion criteria included adults aged 18 years or older, diagnosed with solid tumors, who were undergoing active treatment and had completed a minimum of two chemotherapy cycles. Participants who had experiences of nutrition impact symptoms, were capable of articulating their experiences, and voluntarily consented to participate were additional inclusion criteria. The principal investigator, with the assistance of a staff nurse from the cancer center, conducted the participant selection. In addition to the inclusion criteria, participants were further selected using a heterogeneous sampling

technique (maximum variation technique), which incorporated cases based on cancer type, sex, residence (including those residing in support organizations), disease stage, chemotherapy regimen type, and treatment setting (outpatient and inpatient). This approach facilitates the collection of rich data with diverse perspectives and interpretations of the phenomenon across different contexts. Informed consent was obtained from all participants prior to their involvement in the study. The final sample size was determined by the researcher's assessment of information saturation during interviews and was confirmed during analysis when no additional codes emerged (thematic saturation) [33]. After reaching thematic saturation with 26 interviews, the principal investigator conducted two additional interviews (verification cases). As a result, data collection was ultimately terminated, and saturation was confirmed when the analysis of the verification cases aligned with the existing codes, rather than introducing new insights [34]. A systematic review indicated that saturation is typically achieved within 9–17 interviews and 4–8 focus group discussions in studies involving a relatively homogeneous study population (within a single country) and had narrowly defined objectives [35].

## Reflexive statements

The principal investigator (PI) is a doctoral student in clinical nutrition, possessing a background in clinical nursing and public health nutrition. He has completed an advanced course in qualitative research methods, which included a project component. This study constitutes a part of his dissertation, which utilizes a mixed-methods design. Although the PI is not currently working in the study area, he has exposure of supervising nursing students at the oncology center of a hospital located in a different region of the country.

Furthermore, among the five co-authors involved in this study, four have previously co-authored publications in qualitative research. Additionally, BSE is a public health nutritionist with significant expertise in qualitative research and has authored numerous articles in this field. In addition to the qualitative research credentials, MA is a senior clinical oncologist with extensive experience in treating patients with various cancer types. AJ is a cancer epidemiologist, while AA is a community nutritionist; neither has direct clinical exposure to cancer patients. ZD is a dietitian who instructs students in dietetics and clinical nutrition and also provides treatment to patients with diverse health problems. The authors declare that they have no family members who are cancer survivors, aside from the clinical exposure experienced by some of the authors.

## Bracketing

In this study, the practice of bracketing involves the PI explicitly recognizing his preexisting beliefs regarding nutrition impact symptoms, which are informed by prior clinical experience in a different region of the country. These personal perspectives and preconceived notions about nutritional symptoms are systematically brainstormed and recorded in a reflexive journal (notebook), primarily before starting data analysis. During the analysis phase, this journal is frequently reviewed to maintain an open mental state and to ensure that the findings accurately reflect the participants' experiences [31,36]. For instance, the PI presumed that fruits and vegetables would be the most preferred food groups among all cancer patients undergoing chemotherapy. Nevertheless, this was not universally the case, as some patients expressed a preference for animal-based foods instead.

## Data collection

The principal investigator collected data through face-to-face, in-depth interviews using a semi-structured interview guide and a tape recorder. The guide was developed after determining the study's purpose and research questions, identifying eligible participants, and reviewing relevant literature on chemotherapy-induced nutrition impact symptoms [37]. The interview guide was initially developed in English and subsequently translated into Amharic by the principal investigator. It was then back-translated into English by a doctoral colleague proficient in both languages. This process ensured the preservation of meaning, cultural equivalence, and contextual accuracy of the question items and probes. Any discrepancies

identified in some items were resolved through discussion. The interview guide included background information and cancer-related characteristics, as well as open-ended questions about participants' experiences of symptoms, their impact on daily functioning, food preferences, coping strategies, and perceptions of nutritional care provided in the hospital (Table 1). The interview guide was pre-tested on three patients, resulting in the removal of redundant items and modifications to both the wording and the sequence of questions. Interviews were conducted following the physician's visits with patients deemed fit for chemotherapy. For those receiving outpatient chemotherapy, the interview took place after the completion of their treatment, in the head nurse's office, to ensure a more comfortable environment. Interviews with hospitalized patients were conducted in the afternoon after attendants left the ward, as the patients were often fatigued; relocating them to a separate room was not practical. Interviews were audio-recorded, and field notes were taken to document non-verbal expressions. Each interview lasted 30–45 minutes.

Before the interview, the principal investigator explained the study's purpose, potential risks and benefits, especially regarding their financial support expectations, confidentiality, right to withdraw from the study, and the recordings. During the interview, distractions were minimized by turning off mobile phones, choosing quiet times in the wards, and conducting interviews for outpatients in a separate room. Active listening techniques were employed, such as asking follow-up questions, maintaining neutrality regarding patients' experiences, and adapting to emotional responses associated with being a cancer survivor [38].

## Data processing and analysis

The first author initially transcribed the Amharic audio recordings, incorporating details from field notes taken during each interview. During the transcription process, audio recordings of suboptimal quality were addressed by employing bracketed ellipses [...] and timestamp annotations to denote technical issues, such as [00:25:10 – background noise interference]. Additionally, segments of the transcript were flagged for subsequent verification with participants, either during their next chemotherapy cycle or via telephone communication if they were undergoing their final chemotherapy session at the

**Table 1. Summary of key questions and follow-up probes used in the phenomenological study.**

| S. N | Main questions | Probes |
|------|----------------|--------|
| 1. | Could you tell me what symptoms you experienced before your disease/cancer was diagnosed? | **Focus on nutrition impact symptoms:** |
| 2. | What have been the most challenging experiences you've had with food during your chemotherapy? Share me in detail please. | **Focus on nutrition impact symptoms:** Nausea, vomiting, altered taste, loss of appetite, early satiety, fatigue…etc. |
| 3. | How have these symptoms affected your daily life? | Dietary pattern, activities of daily livings, physical fitness, job, social life, spirituality, stress/anxiety, sleep pattern |
| 4. | How do you deal with these symptoms in your daily life? | Medication, dietary approaches, taking vitamins, fasting, acceptance, prayer to God, tooth brush Effectiveness of anti-emetics? |
| 5. | What foods have become easier for you to eat during chemotherapy, and why is that? | Liquid/semi solid, fruits, vegetables, spices, milk, tubers…etc. |
| 6. | What foods have become hard for you to eat during chemotherapy, and why is that? | Liquid/semi solid, fruits, vegetables, spices, milk, tubers…etc. |
| 7. | Are you able to easily access the foods you prefer to eat? | Economic and physical access, family and social support |
| 8. | What did you think about the nutrition support and meals you received in the hospital? | Tailored dietary counselling, booklet, nutritional assessment, high protein diet provision, high-energy nutritional products |
| 9. | Based on your experience, what recommendations do you have to improve nutritional care for cancer patients? | Nutritional information/counselling, flyers, quality of food in hospital, symptom management… |

time of the interview. The verbatim transcriptions were then translated into English using a contextual translation approach to preserve the intended meaning, cultural nuances, and socio-economic contexts. The researcher brainstormed, documented, and regularly reviewed preconceived assumptions to minimize their influence during data analysis (bracketing). The translated transcripts underwent iterative review, enabling the researcher to familiarize with the textual content and contextual nuances. Through repeated engagement with the data, relevant information was systematically identified. Subsequently, empirical indicators were coded using an interpretative-focused coding strategy. This approach considered the phenomenon under investigation, the research purpose, and the background characteristics of the participants. Interpretative Phenomenological Analysis (IPA) follows hermeneutic principles and views the researcher as an interpreter of the firsthand experiences shared by participants. IPA can be seen as both a research approach and a method for data analysis [30,31,39]. An inductive thematic analysis approach was used to identify emerging themes from the excerpts. To ensure the trustworthiness of the findings, the transcript was independently coded by both the principal investigator and a doctoral student. Furthermore, an audit trail of the entire research process, including the coding procedures, was verified by one of the co-authors (BSE), who is an expert in qualitative research. Relationships among the themes were examined, and a conceptual framework was developed to clarify the theoretical connections between them. The data analysis process was facilitated using MAXQDA24 software, which generated coding statistics. Both categories and major themes were visually represented using diagrams [31]. To ensure methodological rigor and transparency in the research process, the entire content of the study is organized according to the Standards for Reporting Qualitative Research (SRQR) checklist, which consists of 21 items (S1 Table) [40].

### Trustworthiness and rigor

*Credibility:* The interview guide was pilot-tested on three participants, with corrections made based on their feedback. To establish rapport with participants, the study's objectives were clearly communicated, and assurances were provided regarding confidentiality, the absence of direct incentives, and potential risks. Prolonged engagement in oncology clinics was undertaken to gain a comprehensive understanding of the setting and patient interactions. Prior to data analysis, audio recordings were reviewed multiple times, and the transcripts were meticulously examined (immersion). In instances of poor audio quality and ambiguous responses, follow-up phone calls were made, and participants were met twice during their subsequent treatment sessions. Additionally, the researcher included reflexive statements to provide readers with context for interpreting the findings.

*Dependability:* Two doctoral students independently coded the transcripts. Subsequently, the codes were systematically compared by transcript sections. Consensus was achieved for the majority of the codes, although some discrepancies in word choice were noted. In instances of disagreement, a discussion meeting was convened, during which each coder articulated their rationale and collaboratively reviewed the raw data. In cases of persistent disagreement, one of the co-authors (BSE), a senior qualitative researcher, was consulted for resolution.

Furthermore, a detailed description of all research steps, analytical decisions, and interpretive choices was documented in the manuscript. Additionally, the co-author, BSE, evaluated the interview guide, data collection process, analysis, and findings. As a result, discussions were held, and suggestions for improvement, including quote presentation, theme development, and discussion of findings, were incorporated (audit trail).

*Transferability:* To enhance the transferability, we described the research setting, participant characteristics, and sampling technique in detail. Participants were purposively selected using heterogeneous sampling technique from both sexes, various cancer types, different age groups, and treatment settings to gain a rich understanding of their symptom experiences. Thick descriptions of procedures and findings were also presented to enhance the application of the study to different study subjects and contexts.

*Confirmability*: A reflexivity journal was maintained throughout the study to acknowledge the researcher's previous experience in a different cancer center. Data analysis was conducted by the principal investigators and another doctoral student

in the same field. All interpretations were supported by direct participant quotes and observational evidence from field notes. The audit trail implemented by the qualitative research expert was employed to ensure the confirmability of the findings.

## Ethics statement

This research constitutes a component of a PhD thesis, and ethical approval was obtained from the Institutional Review Board (IRB) of the College of Natural and Computational Sciences, Addis Ababa University (Ref. No. CNCSDO/486/2016/2024), as well as from St. Paul Hospital Millennium Medical College (Ref. No. Pm 23/461). Written informed consent was obtained from all study participants following a comprehensive explanation of the study's purpose, procedures, duration, risks, and benefits. Participant identifiers were omitted in the presentation of supportive quotes.

# Results

## Background characteristics

The participants' age ranges from 24 to 70 years (mean age, 43 ± 11 years). The majority were female (80.8%), married (65.4%), attended primary school (34.6%), housemaids (46.2%), lived in Addis Ababa (38.5%), and diagnosed with breast cancer (30.8%) (Table 2).

**Table 2. Background characteristics of study participants.**

| Variables | Category | Frequency (%) |
|---|---|---|
| Age (year) | < 40 | 9 (34.6%) |
| | 40–50 | 12 (46.2%) |
| | > 50 | 7 (26.9%) |
| Sex | Female | 21 (80.8%) |
| | Male | 5 (19.2%) |
| Residence | Addis Ababa | 10 (38.5%) |
| | Out of Addis Ababa | 16 (61.5%) |
| Level of education | No formal education | 7 (26.9%) |
| | Primary school | 9 (34.6%) |
| | Secondary school | 4 (15.4%) |
| | College and above | 6 (23.1%) |
| Occupation | Housemaid | 12 (46.2%) |
| | Daily labor | 3 (11.5%) |
| | Merchant | 3 (11.5%) |
| | Other | 7 (26.9%) |
| Oncologic setting | Inpatient | 14 (53.8%) |
| | Outpatient | 12 (46.2%) |
| Type of cancer | Breast cancer | 8 (30.8%) |
| | Cervical cancer | 5 (19.2%) |
| | Colon and ano-rectal cancer | 5 (19.2%) |
| | Sarcoma | 3 (11.5%) |
| | Thymoma | 2 (7.7%) |
| | Others | 3 (11.5%) |
| Frequency of chemotherapy | Every 3 weeks | 16 (61.5%) |
| | Every 2 weeks | 10 (38.5%) |

### Lived experiences of chemotherapy induced nutrition impact symptoms

Initially, a set of 15 codes was identified that represented the lived experiences of chemotherapy recipients. These codes were clustered into six categories (subthemes) and further organized into three overarching themes. Categories represent collections of first-level codes [31]. The three superordinate themes were: 1) Symptom burden and coping, 2) Individualized food choices, and 3) Unmet nutritional support needs. Individualized food choice was the most frequently occurring code and ultimately emerged as an independent theme (Fig 1). Under the theme of symptom burden and coping, three categories emerged, including temporal and individual variations of symptoms, the burden on quality of life, and symptom relief methods. Under the theme of unmet nutritional support needs, three categories were identified: nutritional misconceptions, economic barriers to food access, and hospital food experiences (Fig 2).

### Theme 1: Symptom burden and coping

**Category 1.1. Temporal and individual variations.** It was revealed that for most patients, nutrition impact symptoms occur and worsen during the first few days after chemotherapy and within the first three cycles. However, some patients may begin experiencing these symptoms during the middle or fifth cycle, while having relatively mild symptoms earlier. Female participants described early-stage symptoms following chemotherapy as unrelenting, comparing them to the nausea and vomiting often experienced during the first trimester of pregnancy. They reported a strong dislike for all foods, including their previous favorites.

*"In the beginning, it was very difficult. For 3 to 4 days after chemotherapy, I was completely not myself. At home, I sit far from the kitchen to avoid the smell. Thanks to God, I survived and can talk like this now."* (Female, 45 years old, breast cancer)

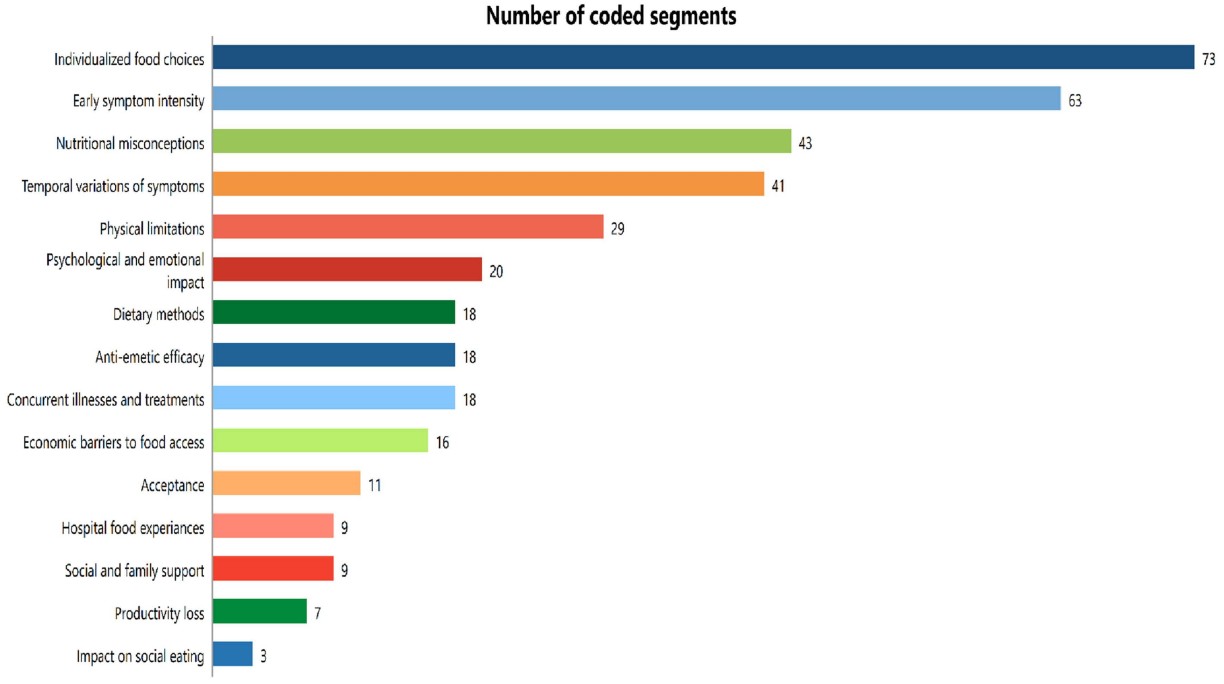

**Fig 1. Summary of codes generated from MAXQDA software.**

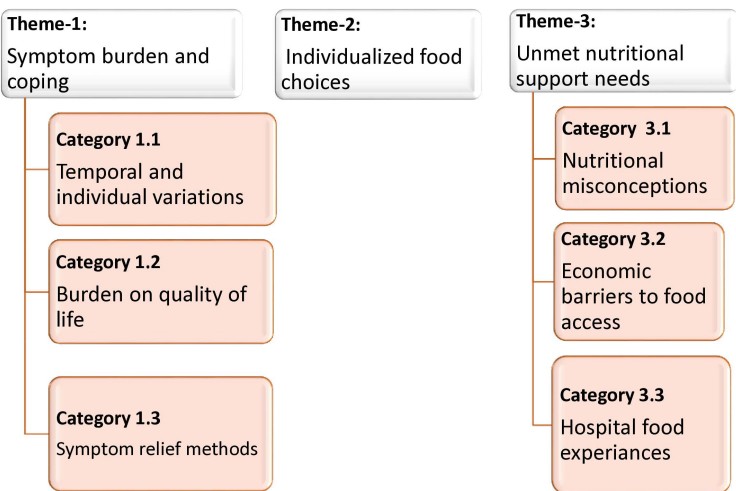

**Fig 2. Schematic presentation of themes and categories.**

Additionally, in most patients, symptoms gradually subsided, allowing them to resume normal eating habits. However, for a few patients, these symptoms persisted or worsened throughout their chemotherapy cycles.

> *"Three days after chemo, I started eating injera (Ethiopian flat bread) with hot pepper and drinking milk, and I gradually began to feel better."* (40 years old, cervical cancer)

> *"I experience vomiting during all six rounds of treatment, and I can't seem to adapt to it."* (Female, 55 year old, colorectal cancer)

Given that chemotherapy is administered every 2–3 weeks, patients often do not have sufficient time to eat adequately between treatments. In majority of cases, this pattern of symptoms typically leads to weight loss; although in rare instances, weight gain has been reported.

> *"By the time I feel better and start eating, it's already time for my next round of chemotherapy."* (Female, 38 year old, colon cancer)

Pre-existing or newly developed gastritis was frequently cited as an underlying factor exacerbating the severity of nutrition impact symptoms.

> *"I developed these irritating symptoms of gastritis after starting chemotherapy. Previously, I used to eat red stews, but I have completely stopped since beginning treatment."* (Female, 45 year old, sarcoma)

Additionally, a few patients living with HIV/AIDS who were undergoing antiretroviral therapy experienced an increased burden from chemotherapy side effects.

> *"I stopped taking my HIV medication while I begin on chemotherapy because it was upsetting my stomach and making me feel very tired."* (Cervical cancer, 5th chemotherapy cycle)

**Category 1.2. Burden on quality of life.** The study revealed that symptoms restricted physical mobility and impaired daily activities, including the ability to attend places of worship. Consequently, patients often experience job loss or are compelled to take annual leave, leading to financial difficulties for both the patients and their families. The economic burden is further increased by additional healthcare costs associated with hospital admissions resulting from the severity of symptoms.

*"After chemotherapy, I usually feel very tired, and sometimes I can't stand up straight; while I am trying to walk I feels like I'm drunk from alcohol."* (Cervical cancer, 5th cycle chemotherapy)

*"I stopped my job after starting chemotherapy and took annual leave from my workplace."* (Female, 35 year old, thymoma)

Moreover, the consequences of nutrition impact symptoms extend to the psychological, emotional, and social lives of participants. Particularly, the fear of experiencing gastrointestinal symptoms affects patients' ability to enjoy mealtimes with friends and family, especially when dining outside their homes.

*"I often ate alone, either before or after family mealtimes, due to my fear of getting sick while eating with them."* (Female, 4th chemo cycle)

*"I prefer to be in a quiet room and don't want to talk to anyone." (Female, 45 year old, sarcoma)*

**Category 1.3. Symptom relief methods.** To alleviate symptoms, patients employed both pharmacological and dietary approaches, which could be either professionally prescribed or self-initiated. While most patients experienced relief from antiemetic medications, many reported no significant improvement. Some participants utilized dietary strategies informed by healthcare professionals, discussions with peers, and online nutrition resources. Economic support from charitable organizations and assistance from family members played a pivotal role in symptom management. Additionally, many patients accepted their condition as an unavoidable side effect of chemotherapy and sought spiritual support as a coping strategy.

*"The anti-vomit medication I took also throw up with the vomit and did not help me, so I stopped taking it."* (Male, 35 year old, sarcoma)

*"I prefer liquids such as milk, gruel, oat and barley soup, and vegetable soups. I also use ENSURE powdered milk that I bought from the pharmacy, and it helps me a lot. I am now able to walk longer distances."* (40 years old, cervical cancer)

### Theme 2: Individualized food choice

Individualized food choice emerged as both a first-level code and an independent theme. It is defined as the selection of foods based on specific needs, preferences, and health goals. This study revealed that food preferences and aversions vary significantly among individuals, even among patients with the same diagnosis undergoing similar treatments. For example, while many patients expressed a preference for fruits, vegetables, cereals (such as 'injera'), and legumes, others reported a strong dislike for these same food groups. Most patients exhibited sensitivity to spices; however, some incorporated spicy foods and condiments, particularly hot peppers, as appetizers during periods of nutritional challenge.

A substantial number of patients prefer boiled eggs and potatoes, while others reject these items. Some have started consuming foods like milk, following healthcare recommendations. However, while some adapted to dietary advice, others experienced nausea and vomiting. Additionally, patients in support organizations often face restrictions due to a fixed

menu. Although they appreciate the logistical support, the monotonous diet does not effectively alleviate the nutritional impacts of chemotherapy.

*"I prefer foods that have a spicy taste, like sauce, pepper, and boiled potatoes."* (30 year old, breast cancer).

*"I used to like oranges and other fruits, but now I can't even stand the thought of them. I hate all fruits and dislike the smell of cooked eggs."* (Female, 47 year old, small bowel cancer)

**Theme 3: Unmet nutritional support needs**

**Category 3.1. Nutritional misconception and disinformation.** Nutritional misconceptions and disinformation refer to false or misleading information about food, diet, and nutrition related to cancer or its treatments from healthcare providers or informal sources. Participants were confused by dietary advice due to unclear information and conflicting recommendations from healthcare professionals. Many relied on personal experiences or family and peer advice rather than formal guidance, citing inconsistencies in recommendations for foods like fruits, vegetables, and dairy products. They emphasized the need for personalized support beyond written materials to navigate dietary choices effectively. Participants face challenges in reconciling dietary advices with personal preferences and community-sourced nutrition information.

*"They told me to avoid all packaged juices, and one doctor also advised me to stay away from bananas. However, they said I could have oranges and milk. Previously, one doctor told me to avoid milk, but another said it was fine to drink it. I also heard from someone nearby that eating eggs is okay."* (41 year old, breast cancer)

**Category 3.2. Economic barriers to food access.** Limited finances force patients to prioritize medication and transportation over food. This creates a vicious cycle in which poor nutrition affects treatment eligibility. Social support systems are insufficient to meet the comprehensive nutritional needs of cancer patients.

*"If I don't eat well, I won't be fit for chemotherapy. I'm also worried about how much it costs to get around. It's all a bit tricky. I'm thinking about using some of the money I usually spend on transportation to buy food and see if that makes a difference."* (41 year old, breast cancer)

**Category 3.3. Hospital food experiences.** It is understood that there is a general dislike for the hospital's food offerings, which many found unsuitable for their dietary needs as cancer patients. The lack of appealing and nutritious options, such as milk, fruits, and gruel, exacerbated their discomfort and dissatisfaction with hospital meals.

*"I have never tried the food provided by the hospital because I don't think it is suitable for cancer patients. I tasted it once, and I believe it is not good even for healthy individuals."* (50 year old, cervical cancer).

**Interpretive synthesis of findings.** The findings indicate that poor symptom management, a lack of patient-centered nutritional support, and financial challenges significantly impacted patients' physical, social, psychological, emotional, and spiritual well-being (Fig 3).

## Discussion

All cancer treatment modalities, including surgical intervention, radiotherapy, and chemotherapy, can substantially alter patients' usual dietary patterns. This study employed interpretative phenomenological analysis to investigate the lived experiences of nutrition impact symptoms within the context of resource-limited environments. The findings identified six

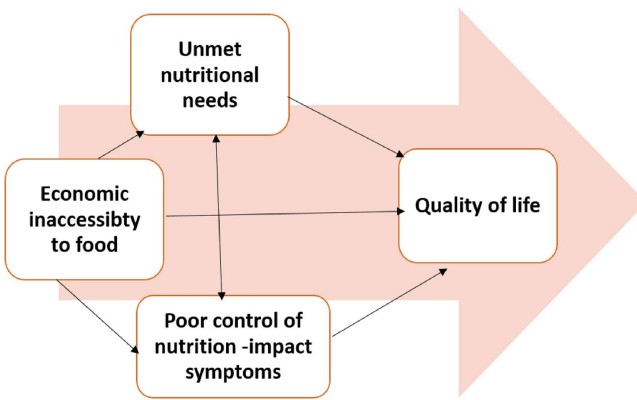

**Fig 3. Conceptual model derived from interpretations of findings.**

categories (subthemes) that describe these experiences: temporal and individual variation of symptoms, burden on quality of life, symptom relief methods, individualized food choice, nutritional misconceptions, economic barriers to food access, and hospital food experiences. The discussion is organized based on the subthemes derived from first-level codes.

This study revealed a subtheme of temporal and individual variation in symptom patterns. In most cases, symptoms begin and worsen within a few days after treatment. However, the exact onset and progression of symptoms vary significantly among individuals. This period is often described as disastrous and intolerable, sometimes necessitating hospitalization and resuscitation between chemotherapy cycles. Most patients endure these challenging times confined to bed and isolated from routine activities. Regarding the temporal variations, some patients experience a gradual subsidence of symptoms, whereas others encounter persistent recurrence at each chemotherapy cycle. Additionally, in some patients, symptoms may emerge at mid-cycle or worsen throughout the treatment period. These variations are likely influenced by factors such as patient age, gender, chemotherapeutic agents, administration frequency, anxiety levels, and prior symptom experiences. Consistent with this observation, chemotherapy-induced nausea and vomiting are categorized into acute, delayed, anticipatory, breakthrough, and refractory types to reflect temporal variations in symptom onset and pattern [41]. Similarly, additional studies have reported individual differences related to the timing of onset, severity, and cyclical nature of nutrition impact symptoms [18,29,42]. This finding underscores the clinical significance of conducting individualized assessments and interventions for nutrition impact symptoms in oncology centers. Additionally, special attention should be paid during the initial days of chemotherapy.

The profound impact of nutrition impact symptoms on quality of life is another subtheme that emerged from this study. This impact is characterized by restricted physical mobility, negative emotional responses, impaired social eating experiences, spiritual visits, and reduced productivity, which can lead to increased economic dependency on family or other social support. Consistent with other studies, the psychological and emotional consequences manifested through anticipatory vomiting, restlessness, depression, sleep disturbances, and avoidance of family meals exacerbate the nutritional challenges faced by patients undergoing chemotherapy [26,43]. Moreover, a meta-synthesis corroborates these findings, highlighting how symptoms such as appetite swings, dysphagia, loss of taste, oral mucositis, and dry mouth further diminish quality of life during treatment [23,44,45]. This result highlights the need for personalized pharmacological and dietary interventions to address the impact of nutritional symptoms on quality of life, especially in low-resource settings. Additionally, the inclusion of oncology-trained dietitians in cancer centers is crucial. However, in the study setting, dietitians are absent from cancer centers, resulting in a neglect of professional nutritional care in oncology clinics. This situation underscores organizational and policy gaps in integrating nutritional care into cancer centers in Ethiopia. Although Ethiopia has a national palliative care guideline, it focuses primarily on pain management, with little emphasis on cancer-specific

nutritional care [46]. This highlights the need to contextualize international guidelines, such as the European Society for Parenteral and Enteral Nutrition (ESPEN) guidelines, for low-resource settings [47]. The barriers to nutritional care in oncology centers, such as the absence of guidelines and gaps in human resources, can be understood through various hierarchies: poor individual knowledge, lack of family and social support, hospital readiness for nutrition, and policy-level gaps. These factors can be explained within the framework of the socio-ecological model (SEM), which illustrates the interactive effects of personal and environmental influences on the poor integration of nutritional care in cancer centers [48]. Therefore, training human resources (dietitians) and developing guidelines at the policy level, implementing nutritional care strategies in hospitals, providing nutritional supplements at the hospital level, and offering tailored dietary counseling to patients at the individual level may mitigate the impact of nutrition impact symptoms on quality of life.

Moreover, symptom relief methods is another subtheme identified from this study. Symptom relief or coping methods includes the use of anti-emetic medication (commonly Plasil), dietary methods, psychosocial support, and spiritual practices. However, the effectiveness of anti-emetics varies between patients. Participants also employed self-preferred dietary methods to manage symptoms based on nutritional information obtained from healthcare providers, informal discussions, and occasional Internet searches. These findings align with those of a study conducted in Saudi Arabia, which reported that patients undergoing chemotherapy continued to experience nausea and vomiting despite receiving antiemetic treatment [49]. On the other hand, patients who experienced symptom relief during chemotherapy attributed this to the premedication and fluids administered in the hospital, which they believed helped mitigate the nutrition impact symptoms. Concurrently, the participants identified psychosocial support from family and charity organizations, acceptance of their condition, and prayer as additional coping strategies [22,50]. These findings have implications for clinical practice; they underscore the importance of using potent anti-emetics, providing patient-centered dietary counseling, and establishing or strengthening support groups for patients. A meta-synthesis further emphasized that supportive relationships between patients' social networks (including those with cancer) and healthcare professionals are particularly crucial in coping with adverse symptoms [51].

This study demonstrates that individualized food choice was the most frequently occurring code, as well as an independent theme synthesized from the transcript. Notably, patients with similar cancer types and those undergoing similar chemotherapy regimens may exhibit different dietary choices. For example, while most participants preferred fruits and vegetables during treatment, a significant proportion also favored tubers, eggs, dairy products, and other animal-based foods. Additionally, while some patients included spices and condiments in their diets, others exhibited heightened sensitivity to these ingredients, which frequently aggravated pre-existing or newly developed gastritis. Studies conducted in Ethiopia have shown that food choices are influenced by various factors including age, religion, food prices, educational status, availability and quality of food, market access, and nutritional literacy [52–54]. In cancer patients, disease characteristics and treatments shape dietary needs and preferences. Consistently, a qualitative study on colorectal cancer patients found that dietary decisions are influenced by foods' perceived effectiveness in reducing treatment side effects [55]. Another study highlighted dietary recommendations aimed at cancer prevention may not align with nutritional guidelines for individuals undergoing cancer treatment [56]. Therefore, emphasizing fruits and vegetables alone may not yield similar benefits post-diagnosis or during cancer treatment. It emphasizes the need for patient-centered dietary counseling in oncology settings. Scientific recommendations should be based on informed decision-making, considering factors such as physical accessibility, economic feasibility, and cultural acceptance.

Furthermore, nutritional misconceptions among patients and health care providers were an important subtheme identified in this study. This misconception was evidenced by observations of dietary practices that contradicted the international clinical nutrition guidelines for patients with cancer. One study showed that many cancer patients understand the importance of optimal nutrition for their health and are eager to make dietary changes [56]. However, feedback obtained from in-depth interviews of this study indicates discrepancies in dietary guidance provided by healthcare professionals, informal community sources, and peer exchanges among patients during chemotherapy sessions. This highlights a

potential knowledge gap among medical practitioners regarding evidence-based nutritional guidelines for cancer patients. It was also recognized that myths and nutritional disinformation exist among community members and non-healthcare professionals in hospitals. On the other hand, a nutritional information booklet was given to patients, however, many needed personalized explanations tailored to their access to foods. A study in Taiwan revealed that the most prevalent unmet need among patients with cancer was nutritional, followed by psychosocial support and patient care, with economic needs also identified [19]. Consistently, a recent qualitative study conducted in Ethiopia identified that healthcare providers' limited nutritional knowledge and inadequate recognition of nutrition as an essential component of routine clinical care were significant barriers to providing nutritional care for adult patients [57].

Within the overarching theme of unmet nutritional needs, the experience of hospital food emerged as a significant sub-theme. Participants noted that the food provided by the hospital was neither palatable nor suitable for cancer patients. They emphasized the need for specialized nutrition, drawing comparisons to orthopedic and maternity wards where more nutritious options, such as gruel, meat, milk, and eggs, are available, which could similarly benefit cancer patients. This highlights the necessity of improving the quality of food offered to hospitalized patients undergoing chemotherapy. Additionally, some patients resorted to purchasing oral nutritional supplements externally, indicating a need for the oncology pharmacy to supply nutritional products to support patients during treatment. Consistently, a hospital-based cross-sectional study conducted with 567 adult patients admitted to public hospitals in Addis Ababa revealed that the overall satisfaction with hospital meal services was 32.5%. The type and taste of the food, along with the behavior of the meal service provider, were significant determinants of patient satisfaction [58]. Similarly, a study in southern Ethiopia reported a 33% satisfaction rate among patients regarding the regular food services provided in the hospital [59]. However, there is a lack of studies using qualitative approaches to investigate the hospital food experience in Ethiopia, particularly in oncology clinics. Therefore, this study provides additional insights into the experiences within cancer wards through a phenomenological design.

Economic inaccessibility to recommended food items is another important subtheme relevant to cancer centers in low-resource settings. It is recognized that a significant number of patients face financial constraints that hinder their ability to adhere to professional dietary advice. Instead, these patients often prioritize the costs of cancer treatments and other logistical expenses over the foods necessary to maintain treatment tolerance. However, inadequate food intake will make patients unfit for chemotherapy, which leads to reappointments that again exacerbate the economic repercussions. In the study hospital, a significant proportion of chemotherapy medications are procured from private pharmacies, and access to radiotherapy is prohibitively expensive. These expenses indirectly affect their attention and purchasing power for nutritious foods commonly recommended for cancer patients. The financial burden faced by cancer patients in Ethiopia, coupled with the inadequacies in cancer treatment facilities, has been consistently reported in various published studies [50,60,61]. Support organizations operating in Addis Ababa offer housing and partial coverage for treatment expenses; however, these services are restricted to women diagnosed with breast or cervical cancer. Overall, the finding has a policy implication that expanding oncology centers to peripheral regions of the country and equipping cancer centers with essential medications and radiotherapy machines could mitigate the logistical challenges encountered by cancer patients. Such enhancements may facilitate patients' access to adequate nutrition, thereby improving their ability to tolerate cancer treatments. Furthermore, the findings underscore the necessity for social protection policies for cancer patients and increased support from both governmental and non-governmental organizations to alleviate the financial burdens experienced by this population.

This study has strengths and limitations. A key strength is its methodology, including face-to-face interviews with diverse participants and audio recordings. The use of interpretative phenomenological analysis enables an understanding of patients' experiences across contexts. Limitations include not incorporating healthcare professionals' perspectives on nutritional care challenges due to the study's phenomenological design, which focuses on patients' experienced nutrition impact symptoms. Participation was limited to Amharic-speaking patients. Social desirability bias may have occurred during patients' sharing of hospital food experiences. Despite efforts to mitigate researcher bias through position statements and reflexivity, unintentional biases may have influenced interviews or data interpretation.

## Conclusions

Patients undergoing chemotherapy experience various nutritional challenges, with symptom patterns and dietary preferences varying on an individual basis. Additionally, inadequate nutritional care within hospital settings, combined with financial difficulties, exacerbates the complexity of managing these symptoms. Therefore, the integration of individualized nutritional interventions alongside socioeconomic support is essential to mitigate the burden of nutrition impact symptoms on quality of life.

## Supporting information

**S1 Table. Standards for Reporting Qualitative Research (SRQR) checklist.**
(DOCX)

**S2 File. Dataset.**
(XLSX)

## Acknowledgments

We express our gratitude to Nutriset and the American Cancer Society for their support in data collection and extend our appreciation to the study participants. We also thank Dr. Diana Cardenas for her constructive comments. This research constitutes a component of the first author's dissertation.

## Author contributions

**Conceptualization:** Awole Seid.

**Formal analysis:** Awole Seid.

**Investigation:** Awole Seid.

**Methodology:** Awole Seid.

**Software:** Awole Seid.

**Supervision:** Zelalem Debebe, Abebe Ayelign, Bilal Shikur Endris, Mathewos Assefa, Ahmedin Jemal.

**Validation:** Bilal Shikur Endris.

**Writing – original draft:** Awole Seid.

**Writing – review & editing:** Zelalem Debebe, Abebe Ayelign, Bilal Shikur Endris, Mathewos Assefa, Ahmedin Jemal.

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
