## [Decision Letter · Decision Letter 0]

13 Oct 2025

Dear Dr. Seid,

We look forward to receiving your revised manuscript.

Kind regards,

Muktar Beshir Ahmed, PhD

Academic Editor

PLOS ONE

Journal Requirements:

3. Please amend the manuscript submission data (via Edit Submission) to include author Diana Cardenas.

Additional Editor Comments:

Both reviewers found the study scientifically interesting and methodologically sound overall, but they have raised several points that need attention before the manuscript can be considered for acceptance.

**Required revisions (must be addressed):**

Clarify key aspects of the methodology as outlined by both reviewers.Address Reviewer 2’s concerns about [e.g., the interpretation of results].Ensure consistency between the results and discussion sections and improve clarity of the main findings.Revise the abstract and conclusion to accurately reflect the study’s scope and limitations.

Reviewers' comments:

Reviewer's Responses to Questions

**Comments to the Author**

1. Is the manuscript technically sound, and do the data support the conclusions?

Reviewer #1: Yes

Reviewer #2: Partly

2. Has the statistical analysis been performed appropriately and rigorously?

Reviewer #1: N/A

Reviewer #2: N/A

3. Have the authors made all data underlying the findings in their manuscript fully available?

Reviewer #1: No

Reviewer #2: Yes

4. Is the manuscript presented in an intelligible fashion and written in standard English?

Reviewer #1: Yes

Reviewer #2: Yes

Reviewer #1: Dear Academic Editor,

I hope this message finds you well. I am writing to provide my comments on the manuscript titled "[Lived Experience of Nutrition Impact Symptoms among Patients Undergoing Chemotherapy in Ethiopia: An Interpretative Phenomenological study]" (Manuscript ID: [PONE-D-25-19003]) that I had the opportunity to review.

I appreciate the opportunity to review this manuscript and hope that my feedback will be helpful to the authors in improving their work. Please let me know if you require any further information or clarification regarding my comments.

Thank you for considering my review

Overall, I found the manuscript to be well-structured and innovative. However, I have some suggestions and comments that I believe could enhance the quality of the work, which are presented below.

Comment #1: The title stated in lines no 1 and 2 stated that “Lived Experience of Nutrition Impact Symptoms among Patients Undergoing Chemotherapy in Ethiopia…” But the concept that you stated in the abstract part in the background section, lines no 32 and 33, is “This study aimed to explore the lived experiences of symptoms that hinder food intake among chemotherapy recipients at a major cancer center in Ethiopia.” Hence, there is a misalignment between the title and the objective. Please revisit it.

Comment #2: In the abstract in the method section, line no 35, you mentioned “…purposive sampling…” This sampling method is the most preferred in qualitative studies, but it has many types of sampling techniques. Please state the specific type of purposive sampling you used in your study.

Comment #3: In the abstract in the method section at line no 35, you stated “… involving 26…” You did not mention how you reached at this level. Please precisely mention such size determination method that you applied.

Comment #4: In the abstract in the method section at lines no 36 to 37, you stated “…. transcribed verbatim in Amharic, and then translated into English”. You used a verbatim approach for transcription, but you did not mention the method that you used for translation, and you also missed it in the method part. Please precisely include this approach here.

Comment #5: The gaps that you what to fill in this study using a qualitative approach, including the paradigm, are sufficiently described. Please include it.

Comment #6: In the methods part in the study participants and selection section at lines no 115 to 117, you stated “….A total of 26 participants were purposively selected based on their experiences with nutrition-impact symptoms, diverse age groups, sexes, cancer types, stages, and treatment regimen…” This indicated that diverse criteria were used to recruit the participants. Hence, it is necessary to indicate the types of purposive sampling you used.

Comment #7: In the methods section from line no 120 to 132, you describe the Reflexivity statement and bracketing.” It is good to understand the position of the researchers about the issue that was investigated and how researchers' bias is introduced into the findings. But only the first author's and one other author's issues related to the topic were described. Hence, it lacks the position of other authors on the issue. Disclosure about the presence of cancer in the families of the authors is essential. Hence, precisely include such and other related events.

Comment #8: In the methods part in the data collection section, from lines no 142 to 146, you describe how you interview the participants. In this part, it is also necessary to include the mechanisms you used to maintain and engage the interest of the participants during the interview time.

Comments #9: In cases where the audio quality was poor or the discussion required clarification, what mechanism did you apply during audio transcription and translation?

Comment #10: In the methods part, you described trustworthiness and rigor assurance mechanisms. It is good to include additional mechanisms that you applied to assure the trustworthiness of your findings (like an iterative process, modifications of study procedure in response to evolving study findings not adequately mentioned, figuring out contradicting ideas, and further verifying them, etc).

Comment #11: The discussion is supported by evidence, but some areas focus heavily on summarizing findings without giving meaning and specification of implications. Therefore, consider elaborating on how the findings challenge or confirm existing theories. This is my major concern, please revisit this in-depth.

Comments # 12: Please follow the qualitative research guideline of Plos one.

Reviewer #2: This is a well-conceived and timely qualitative study exploring the lived experiences of nutrition-impact symptoms (NIS) among Ethiopian cancer patients undergoing chemotherapy. However, there are areas where clarity, depth, and alignment with reporting standards could be strengthened. Detailed comments and suggestions are attached separately.

**Do you want your identity to be public for this peer review?** For information about this choice, including consent withdrawal, please see our Privacy Policy

Reviewer #1: **Yes: ** Yinager Workineh

Reviewer #2: No

---

## [Author Response · Author response to Decision Letter 1]

26 Oct 2025

Response to the Editor and Reviewers

Firstly, on behalf of the research team, I would like to thank the editor and reviewers for their valuable time and constructive comments on our manuscript. Their feedback will enhance the scientific rigor of the work. Below, I provide detailed responses to each comment. All modifications are highlighted in red color with the “revised manuscript with track changes”.

Response to the editor

1. A response to reviewers, a revised manuscript with track changes, and a clean version of the manuscript are uploaded.

2. The manuscript has been prepared in accordance with PLOS ONE's style requirements. Its organization has been verified using the Standards for Reporting Qualitative Research (SRQR) checklist, as recommended by the journal. The checklist is also uploaded to the submission system as supporting information.

3. The data sharing statement has been revised to: “All relevant data are within the manuscript and its supporting information files.” The dataset is exported directly from MAXQDA software to an Excel sheet, which is uploaded as a supporting information file. I believe the Excel format is clear and straightforward for potential replication.

4. Dr. Diana Cardenas considers her contribution limited to being recognized as a co-author and requests her name to be included in the acknowledgment section. Consequently, her name has been moved to the acknowledgment. I apologize for any inconvenience this may have caused.

5. The reviewers did not recommend citing specific previously published works.

Response to reviewer – 1

Comment #1: The title stated in lines no 1 and 2 stated that “Lived Experience of Nutrition Impact Symptoms among Patients Undergoing Chemotherapy in Ethiopia…” But the concept that you stated in the abstract part in the background section, lines no 32 and 33, is “This study aimed to explore the lived experiences of symptoms that hinder food intake among chemotherapy recipients at a major cancer center in Ethiopia.” Hence, there is a misalignment between the title and the objective. Please revisit it.

Response: Accepted. Although, nutrition impact symptoms refers the symptoms that hinder food intake, the comment to use consistent terms throughout the manuscript is acceptable and incorporated. Page 2, line 31-32.

Comment #2: In the abstract in the method section, line no 35, you mentioned “…purposive sampling…” This sampling method is the most preferred in qualitative studies, but it has many types of sampling techniques. Please state the specific type of purposive sampling you used in your study.

Response: Accepted. The specific sampling technique used in the study was a heterogeneous sampling (maximum variation sampling) technique where we mixed patients by gender, cancer type, treatment setting (outpatient versus inpatient), the type of therapeutic agents, and the number of chemotherapy cycles the patient was taking. The specific sampling technique is incorporated in both the abstract and method sections. Page 2, lines 36 and page 5, lines 132 - 137, respectively.

Comment #3: In the abstract in the method section at line no 35, you stated “… involving 26…” You did not mention how you reached at this level. Please precisely mention such size determination method that you applied.

Response: Accepted. Notably, saturation in thematic analysis is a debated topic among qualitative researchers. In our study, the sample size was determined when no new information emerged during data collection (data saturation) and no new codes or themes appeared during analysis (thematic saturation). After completing data analysis with 26 samples, we collected two additional verification cases to capture any new information or codes that might arise. At this point, saturation was declared due to the absence of new responses from the interviews, and the data analysis confirmed existing codes rather than revealing new ones (thematic saturation). Consistently, the sample size in our study is somewhat larger than that of a systematic review, which indicated that saturation in homogeneous study populations (single country) is often reached within 9 to 17 interviews.

Based on the comment, a precise statement on how the sample size was determined using data and thematic saturation has been included in the abstract (lines 35-36). Additionally, detailed descriptions of the procedures have been added to the methods section on page 6, lines 138-146.

Comment #4: In the abstract in the method section at lines no 36 to 37, you stated “…. transcribed verbatim in Amharic, and then translated into English”. You used a verbatim approach for transcription, but you did not mention the method that you used for translation, and you also missed it in the method part. Please precisely include this approach here.

Response: Accepted. As this study is an interpretative phenomenological analysis, we utilized contextual translation. This approach is essential because the wording, as well as the patients' disease status and socio-economic context, significantly contribute to the interpretation of their symptom experiences. Contextual translation preserves the intended message and socio-cultural nuances. The type of translation is mentioned in the abstract (line 38) and presented in detail in the methods section (Page 9, lines 210-212).

Comment #5: The gaps that you what to fill in this study using a qualitative approach, including the paradigm, are sufficiently described. Please include it.

Response: I take this comment as “not sufficiently described.” The justification for the qualitative approach and constructivist paradigm of this study is detailed on page 4, lines 94-100.

Comment #6: In the methods part in the study participants and selection section at lines no 115 to 117, you stated “….A total of 26 participants were purposively selected based on their experiences with nutrition-impact symptoms, diverse age groups, sexes, cancer types, stages, and treatment regimen…” This indicated that diverse criteria were used to recruit the participants. Hence, it is necessary to indicate the types of purposive sampling you used.

Response: Accepted. This question is addressed in comment #2. The heterogeneous sampling technique defines the specific method we used in our study. Detailed information about this sampling technique can be found on page 5, lines 132–137.

Comment #7: In the methods section from line no 120 to 132, you describe the Reflexivity statement and bracketing.” It is good to understand the position of the researchers about the issue that was investigated and how researchers' bias is introduced into the findings. But only the first author's and one other author's issues related to the topic were described. Hence, it lacks the position of other authors on the issue. Disclosure about the presence of cancer in the families of the authors is essential. Hence, precisely include such and other related events.

Response: Accepted. In this section, the second reviewer also suggested adding the professional background and previous experience in qualitative research of the principal investigator and co-authors. A statement detailing this professional and qualitative research experience is included in the reflexive statement on page 6, lines 148-163.

Comment #8: In the methods part in the data collection section, from lines no 142 to 146, you describe how you interview the participants. In this part, it is also necessary to include the mechanisms you used to maintain and engage the interest of the participants during the interview time.

Response: Accepted. During the interview, distractions were minimized by turning off mobile phones, selecting quiet times in the wards, and conducting interviews for patients undergoing outpatient chemotherapy in a separate room (the head nurse’s office). Active listening, follow-up questions, maintaining neutrality regarding patients’ experiences, and adapting to emotional responses were used during the interview. Additional details can be found in the methods section on page 8, lines 198-202.

Comments #9: In cases where the audio quality was poor or the discussion required clarification, what mechanism did you apply during audio transcription and translation?

Response: During the transcription process, audio recordings of suboptimal quality were addressed by employing bracketed ellipses [...] and timestamp annotations to denote technical issues, such as [00:25:10 - background noise interference]. Additionally, segments of the transcript were flagged for subsequent verification with participants, either during their next chemotherapy cycle or via telephone communication if they were undergoing their final chemotherapy session at the time of the interview.

While not frequent, the inclusion of statements detailing procedures for addressing poor quality audio records, primarily background noise interference in the inpatient wards, is acceptable and has been incorporated into the data analysis section. Page 9, lines 205 – 210.

Comment #10: In the methods part, you described trustworthiness and rigor assurance mechanisms. It is good to include additional mechanisms that you applied to assure the trustworthiness of your findings (like an iterative process, modifications of study procedure in response to evolving study findings not adequately mentioned, figuring out contradicting ideas, and further verifying them, etc).

Response: Accepted. Familiarization with the content through an iterative process (repeated listening and reading) is reported in the methodology (page 11, lines 238 - 242). However, in this study, modification of the study procedure was not made in response to study findings. Importantly, the interview guide was pre-tested on three patients, resulting in the removal of redundant items and modifications to both the wording and the sequence of questions. The statement on pre-test of interview guide is stated in the methodology (page 8, lines 187 - 189).

Notably, unlike phenomenological studies, modifying study procedures based on ongoing analysis and findings is common in grounded theory and, to some extent, in ethnographic studies.

Comment #10: In the methods part, you described trustworthiness and rigor assurance mechanisms. It is good to include additional mechanisms that you applied to assure the trustworthiness of your findings (like an iterative process, modifications of study procedure in response to evolving study findings not adequately mentioned, figuring out contradicting ideas, and further verifying them, etc).

Response: Accepted. Some rigor assurance methods that were implemented but not detailed, such as prolonged engagement, immersion in data, and an audit trail, are now included in the trustworthiness section. Regarding dependability, in addition to peer debriefing with a doctoral student colleague, the overall study process, from the interview guide to manuscript preparation, was thoroughly evaluated by one of the co-authors (BSE), a qualitative research expert (Audit Trail). However, as explained above, the study procedure was not modified in response to evolving findings. I also understand that all the trustworthiness and rigor assurance mechanisms may not apply to all study designs and contexts. The expansion statement on trustworthiness and rigor assurance methods is included on page 11, lines 238 – 241 and lines 251 – 254.

Furthermore, during the coding process, we observed a contradiction in participants' food choices. Some preferred fruits and vegetables during chemotherapy, while others disliked them. Similarly, some used spices to boost appetite, whereas others avoided them due to stomach irritation. This lack of a clear pattern led to the development of the code "individualized food choice," which became an independent theme in the final analysis. We did not identify any additional contradictory findings.

Comment #11: The discussion is supported by evidence, but some areas focus heavily on summarizing findings without giving meaning and specification of implications. Therefore, consider elaborating on how the findings challenge or confirm existing theories. This is my major concern, please revisit this in-depth.

Response: Thank you for your comment. It is important to note that the discussion section is organized by sub-themes, with each paragraph addressing one sub-theme at a time. The final lines of each paragraph do not serve as summaries; rather, they highlight the potential implications of specific findings for improving clinical practice, which I believe is a fundamental component of a discussion section.

In response to this general comment, the discussion section has been thoroughly revised and expanded to contextualize the findings within the framework of other studies and existing theories. See pages 18-23 or lines 420-569.

Comments # 12: Please follow the qualitative research guideline of Plos one.

Response: The entire content is organized in accordance with the Standards for Reporting Qualitative Research (SRQR) checklist, which is one of the reporting guidelines recommended by the journal. As evidence, the filled checklist is uploaded as additional material in the submission system. Also, a statement indicating the reporting standard is added in the methods part (Page 9, lines 229-231).

Response to reviewer – 2

Comment on the abstract: Well-written, but maybe include some more details in the results instead of just listing the themes and subthemes, giving some information on what each theme represents.

Response: Accepted. Some details are added in the result section of the abstract, and the conclusion is also rephrased. Lines 40 – 43.

Comment on the introduction: Line 69-72: “In our baseline cross-sectional study, we found that 61.7% of patients in major cancer centers in Ethiopia had at least one NIS before chemotherapy initiation, with expected increases during and after treatment (unpublished data)”. I suggest using published information to support your argument.

Response: Accepted. At the time of submission of this manuscript, the cited manuscript was under review; now it is published and appropriately cited. Reference number 16.

Comment on introduction: Line 74-75: you mention there are “Several qualitative studies have reported that NIS in individuals with cancer receiving treatments is associated with weight loss, poor quality of life, and decreased survival” – but no reference, include references.

Response: It was assumed to serve as a topic sentence introducing a new paragraph. The studies referred to as "several" are listed and explained following this statement. However, if this creates confusion, I rephrased it to better function as a topic sentence for the new paragraph. Page 3, lines 72 – 73.

Comment on introduction: Line 87-88: “However, the findings are limited to British and American contexts, making them less applicable to other cultural settings with different dietary habits”- by this, are we assuming the symptoms vary by setting or by dietary habits? It is good to clarify.

Response: The trend of late cancer diagnosis in Ethiopia, where 80% of cases are identified at an advanced stage, coupled with a limited number of cancer and palliative care centers, ill-equipped oncology facilities, a shortage of nutritionists and dietitians, and economic barriers to accessing treatments and nutritious foods, is expected to increase the burden of nutrition-related symptoms in low-resource settings. The hypothesis is that while the disease remains consistent among countries, however, the environmental conditions may exacerbate the severity of these symptoms. Moreover, dietary habits vary across different settings. To enhance clarity, a summary of these rationales is provided in a commented paragraph. Page 4, lines 86 - 90.

Comment on introduction: Line 89-90: “closer examination of previously published studies lacks a comprehensive understanding of patients' experiences with NIS”- clearly indicate what they are missing and what this study is doing differently to fill the gaps.

Response: Accepted. The gap in previous studies conducted in Ethiopia and other contexts is clearly articulated. See page 4, lines 99 – 113.

Comment: Terminology: Use consistent phrasing for “nutrition-impact symptoms” (sometimes abbreviated, sometimes not).

---

## [Editor Report · Decision Letter 1]

3 Nov 2025

Lived experience of nutrition impact symptoms among patients undergoing chemotherapy in Ethiopia: An interpretative phenomenological analysis

PONE-D-25-19003R1

Dear Dr. Awol Seid,

We’re pleased to inform you that your manuscript has been judged scientifically suitable for publication and will be formally accepted for publication once it meets all outstanding technical requirements.

Within one week, you’ll receive an e-mail detailing the required amendments. When these have been addressed, you’ll receive a formal acceptance letter, and your manuscript will be scheduled for publication.

Kind regards,

Muktar Beshir Ahmed, PhD

Academic Editor

PLOS ONE
---

## [Editor Report · Acceptance letter]

PONE-D-25-19003R1

PLOS ONE

Dear Dr. Seid,

I'm pleased to inform you that your manuscript has been deemed suitable for publication in PLOS ONE. Congratulations! Your manuscript is now being handed over to our production team.

Kind regards,

on behalf of

Professor. Muktar Beshir Ahmed

Academic Editor

PLOS ONE